# Anonymous Homomorphic IBE with Application to Anonymous Aggregation

Michael Clear  and Hitesh Tewari *

School of Computer Science and Statistics, Trinity College Dublin, D02 PN40 Dublin, Ireland
* Correspondence: htewari@tcd.ie

**Abstract:** All anonymous identity-based encryption (IBE) schemes that are group homomorphic (to the best of our knowledge) require knowledge of the identity to compute the homomorphic operation. This paper is motivated by this open problem, namely to construct an anonymous group-homomorphic IBE scheme that does not sacrifice anonymity to perform homomorphic operations. Note that even when strong assumptions, such as indistinguishability obfuscation (iO), are permitted, no schemes are known. We succeed in solving this open problem by assuming iO and the hardness of the DBDH problem over rings (specifically, $Z_{N^2}$ for RSA modulus $N$). We then use the existence of such a scheme to construct an IBE scheme with re-randomizable anonymous encryption keys, which we prove to be IND-ID-RCCA secure. Finally, we use our results to construct identity-based anonymous aggregation protocols.

**Keywords:** identity-based encryption; homomorphic encryption; anonymous aggregation



## 1. Introduction

The problem we tackle in this paper relates to a primitive known as identity-based group homomorphic encryption (IBGHE), which is defined in [1]. Basically, IBGHE is identity-based encryption that is homomorphic for some group operation, and the ciphertext space for every identity forms a group. Moreover, the decryption function is a group homomorphism between the ciphertext group and the plaintext group. GHE has several applications, discussed in [1], and an IBGHE facilitates those applications in an identity-based infrastructure.

It is an open problem to construct an IBGHE that is simultaneously anonymous and homomorphic for addition. There are only two IBGHE schemes that support modular addition to the best of our knowledge, namely the XOR-homomorphic variant of the Cocks IBE scheme in [1] and the more recent IBGHE scheme from [2] that is homomorphic for addition modulo smooth square-free integers. Now, Joye has discovered that the Cocks IBE scheme itself is XOR-homomorphic [3], but the scheme is not an IBGHE since the ciphertext space with the homomorphic operation forms a quasigroup and not a group. Some readers might wonder about schemes that are considered multiplicatively homomorphic, which allow addition in the exponent, and question why we do not classify them as IBGHE schemes for addition. The reason is that the corresponding additive group has exponential order, and decryption can only recover messages using Pollard's lambda algorithm that are less than some polynomial bound, so the *v*alid message space does not form an additive group. Now the two IBGHE schemes supporting modular addition that we are aware of are not anonymous, but there are variants of these schemes that achieve anonymity. However, although such schemes gain anonymity, they lose the homomorphic property. Most usually, we need to know the identity associated with a ciphertext in order to correctly compute the homomorphic operation, and so when the identity is hidden from us, as it is when the scheme is anonymous, we cannot compute the homomorphic operation. Therefore, in a nutshell, the open problem we address in this paper is to construct an IBGHE for addition that is anonymous while retaining the

homomorphic operation. Note that while we have concentrated on GHE, it is important to point out that there are no other additively homomorphic schemes (such as quasigroup homomorphic schemes, such as Cocks, as observed by Joye) that achieve simultaneous anonymity and the ability to carry out the homomorphic operation without knowing the identity associated with a ciphertext. Of course, our focus is not on bounded homomorphisms, such as LWE-based schemes that incorporate noise, but instead on those with an algebraic structure and support for a theoretically unbounded number of operations. One of the reasons we opt for GHE over linearly homomorphic LWE-based schemes is that the former enjoy the desired property of *strong unlinkability*; that is, an evaluated ciphertext is distributed the same as a fresh ciphertext in the view of the key holder (recipient), whereas LWE-based schemes achieve this only by requiring an expensive bootstrapping operation and making a circular security assumption.

### 1.1. Motivation and Applications

Beyond theoretical interest, there are applications that motivate consideration of this open problem. We construct an anonymous IBE using an anonymous IBGHE as a building block. We prove this scheme IND-ID-RCCA secure (note that RCCA is a slight relaxation of CCA2). Our anonymous IBE scheme has two interesting properties. First, it allows one to generate anonymous keys associated with a particular identity. Therefore, an encryptor can encrypt a message using an anonymous key for some unknown recipient. Secondly, such keys can be rerandomized such that the resulting anonymous key is computationally unlinkable to the original anonymous key. This finds an immediate application in anonymous aggregation, as we describe below.

Consider the following application scenario. Suppose we have a collection of sensor nodes that collect data and send it to a central server. Suppose the data are numerical measurements, and there are different recipients depending on external factors. Each sensor data encrypts a measurement with the recipient's identity and sends it en route to the central server. It is desirable that ciphertexts that are seen by potential adversaries do not reveal the associated recipient's identity. Along the route there are nodes that function as aggregators that can be authorized independently by each sensor node to aggregate the data coming from that sensor node. If a sensor node give authorization to the aggregator, then the aggregator should be able to aggregate data for the same recipient coming from any of the sensor nodes that have given authorization. Addition (summation) is a common type of aggregation since perhaps only an average measurement is needed by the recipient. To fulfill this application scenario, we need an IBE scheme that is anonymous and homomorphic for addition, where the homomorphic operation can be computed without knowing the recipient's identity.

Consider two senders that produce ciphertexts for the recipient id. Both of them send their respective authorization keys to an aggregator whose identity is $\bar{\text{id}}$, they perform aggregation on the two ciphertexts and send the result on to a second aggregator. The second aggregator should not be able to perform aggregation with the result unless they are given an authorization key from $\bar{\text{id}}$. However, the recipient should be able to decrypt all such ciphertexts intended for them, including the result of the aggregation. Now the issue is that the recipient's identity is hidden from the aggregators. However, the result of their aggregation needs to be decryptable by the recipient id and also "fresh", such that the second aggregator, who may be authorized by the original senders, but not authorized by the first aggregator, should not be able to perform aggregation on the result. We describe our approach to solving this problem below.

### 1.2. Our Results

We present a feasibility result in this work of an additively homomorphic IBGHE that is both anonymous and supports the evaluation of the homomorphic operation without knowing a user's identity. Our construction is based on iO and the hardness of DDH in elliptic curves over $\mathbb{Z}_{N^2}$ where $N$ is an RSA modulus. These are strong assumptions but we make headway on this open problem. Elliptic curves over rings have been less widely

studied; Pailler [4] introduced the types of curves we use in this paper, which are over the ring $\mathbb{Z}_{N^2}$, while Peter et al. [5] describe a specific class of curves that are suitable to instantiate our construction. Furthermore, iO has not been realized from standard assumptions, although there have been several recent advances in constructing iO from quite different approaches under different assumptions, which gives us more confidence that iO exists. To obtain our feasibility result, we first borrow an idea from [6] to leverage obfuscation to map an identity string to a freshly generated public key of some encryption scheme. In fact, abstracting for a moment from the specific construction, we will describe the high-level paradigm. As part of the public parameters, we have an obfuscated program that maps an identity to a public key in some *multi-user* system with public parameters. The public keys in a *multi-user* system share the same set of common public parameters—think of the generator $g$ and modulus $p$ in ElGamal [7] as the common public parameters, except ElGamal is of no use here since it is only multiplicatively homomorphic. Nevertheless, ElGamal serves to illustrate another property that this paradigm requires, namely that the multi-user system supports key privacy where key privacy can be viewed as the analog to anonymity in the identity-based setting; that is, the ciphertexts in the multi-user system do not reveal the public key they are associated with, which is the case in ElGamal. We are using the term multi-user system in a broad sense here, permitting both the case where we have a trusted authority and the case where we do not. In the former, the public parameters are generated by a trusted authority with a backdoor (master secret key) such that the trusted authority can decrypt any ciphertext. In our paradigm, the public parameters of the multi-user system will be generated by the trusted authority of the IBE scheme and published as part of the IBE scheme's public parameters. Therefore, we need the multi-user system to be both key-private and additively homomorphic, where the homomorphic operation can be computed without knowing the public key associated with a ciphertext. The fundamental question is: can we concretely realize a multi-user system that has both key privacy and an additive homomorphism. We can answer this question in the affirmative by using a variant of the Paillier scheme based on elliptic curves over rings that is presented in [5], which is a *multi-user* system supporting homomorphic addition modulo a large semiprime $N$ and for which we can easily show that key privacy holds assuming the hardness of DDH in elliptic curves over $\mathbb{Z}_{N^2}$.

### 1.2.1. Anonymous IBE with Rerandomizable Anonymous Keys

Next, we present an anonymous IBE scheme based on the Boneh–Franklin scheme, which we prove IND-ID-RCCA secure. Our scheme requires an additively homomorphic anonymous IBE scheme as a building block (as described above and which we realize in Section 3). Our anonymous IBE scheme has two interesting properties. First, it allows one to generate anonymous keys associated with a particular identity. Therefore, an encryptor can encrypt a message using an anonymous key for some unknown recipient. Secondly, such keys can be rerandomized such that the resulting anonymous key is computationally unlinkable to the original anonymous key. One of the applications for this scheme is in realizing identity-based anonymous aggregation in Section 5. This is the first IBE scheme that is both anonymous and IND-ID-RCCA secure.

### 1.2.2. Identity-Based Anonymous Aggregation

In an identity-based anonymous aggregation (IBAA) protocol, every identity has an associated secret key derivable by the Trusted Authority with their master secret key. Every identity can issue an authorization key to an aggregator that allows the aggregator to perform aggregation on ciphertexts created by that identity, but for *any* recipient identity. We envisage that, in practice, more complex policies may be used to control authorization, which is beyond the scope of this work. Here we simply model authorization with symmetric keys. Therefore, a symmetric key functions as an authorization key that can be issued to aggregators. For every ciphertext, the encryptor generates a fresh symmetric key $\kappa$ (effectively a session or transport key) and uses it to encrypt the IBE ciphertext that encrypts the message. This symmetric key $\kappa$ is encrypted with the authorization key for the sender

so that any party who is given this key can recover the IBE ciphertext that encrypts the message. However, the recipient must always be able to decrypt a ciphertext intended for them, irrespective of whether it has been given an authorization key (for aggregation) by the encryptor. To solve this problem, the ciphertext also incorporates an IBE encryption of $\kappa$ so that the recipient can recover the IBE ciphertext that encrypts the message. One of the main challenges is in relation to aggregation. It is straightforward for the aggregator to evaluate the homomorphic operation on both IBE ciphertexts without knowing the recipient's identity (anonymous group-homomorphic IBE enables this). However, we must use a fresh symmetric key to encrypt this evaluated IBE ciphertext in order to ensure unlinkability. However, how do we encrypt this fresh key with the recipient's identity without knowledge of the identity so that they can decrypt the result of the aggregation? One solution to this is to use FHE and then rely on bootstrapping for unlinkability, but this requires us to make a circular security assumption, and, furthermore, bootstrapping in the identity-based settings requires strong assumptions, such as iO. Our solution is to use our anonymous IBE scheme with its rerandomizable anonymous keys (described above), and this solves all our problems (including strong unlinkability) while being more efficient than FHE and without the need for a circular security assumption. Furthermore, we rely on the IND-ID-RCCA security to prove a desirable property of *aggregation validity* whereby no party who has not been granted authorization as an aggregator can perform a pre-determined transformation of the plaintext.

## 2. Preliminaries

### 2.1. Notation

A quantity is said to be negligible with respect to some parameter $\lambda$, written $\mathsf{negl}(\lambda)$, if it is asymptotically bounded from above by the reciprocal of all polynomials in $\lambda$.

For a probability distribution $D$, we denote by $x \leftarrow_\$ D$ that $x$ is sampled according to $D$. If $S$ is a set, $y \leftarrow_\$ S$ denotes that $y$ is sampled from $x$ according to the uniform distribution on $S$.

The support of a predicate $f : A \to \{0, 1\}$ for some domain $A$ is denoted by $\mathsf{supp}(f)$, and is defined by the set $\{a \in A : f(a) = 1\}$.

The set of contiguous integers $\{1, \ldots, k\}$ for some $k > 1$ is denoted by $[k]$.

### 2.2. Identity-Based Encryption

**Definition 1.** *An Identity-Based Encryption (IBE) scheme is a tuple of PPT algorithms $(G, K, E, D)$ defined with respect to a message space $\mathcal{M}$, an identity space $\mathcal{I}$ and a ciphertext space $\hat{\mathcal{C}}$ as follows:*

- $\mathsf{G}(1^\lambda)$:
  *On input (in unary) of a security parameter $\lambda$, generates public parameters $\mathsf{PP}$ and a master secret key $\mathsf{MSK}$. Output $(\mathsf{PP}, \mathsf{MSK})$.*

- $\mathsf{K}(\mathsf{MSK}, \mathsf{id})$:
  *On input of the master secret key $\mathsf{MSK}$ and an identity $\mathsf{id} \in \mathcal{I}$: a secret key $\mathsf{sk}_{\mathsf{id}}$ for identity $\mathsf{id}$ is derived and output.*

- $\mathsf{E}(\mathsf{PP}, \mathsf{id}, m)$:
  *On input of public parameters $\mathsf{PP}$, an identity $\mathsf{id} \in \mathcal{I}$, and a message $m \in \mathcal{M}$, a ciphertext $c \in \mathcal{C} \subseteq \hat{\mathcal{C}}$ that encrypts $m$ under identity $\mathsf{id}$ is output.*

- $\mathsf{D}(\mathsf{sk}_{\mathsf{id}}, c)$:
  *On input of a secret key $\mathsf{sk}_{\mathsf{id}}$ for identity $\mathsf{id} \in \mathcal{I}$ and a ciphertext $c \in \hat{\mathcal{C}}$, a $m'$ is output if $c$ is a valid encryption under identity $\mathsf{id}$; otherwise, a failure symbol $\perp$ is output.*

### 2.3. Public-Key GHE

An important subclass of partial homomorphic encryption is the class of public-key encryption schemes that admit a group homomorphism between their ciphertext space and plaintext space. This class corresponds to what is considered "classical" HE [8], where a single group operation is supported, most usually, addition. Gjøsteen [9] examined the

abstract structure of these cryptosystems in terms of groups and characterized their security as relying on the hardness of a subgroup membership problem. Armknecht, Katzenbeisser and Peter [8] rigorously formalized the notion and called it *group homomorphic encryption* (GHE). We recap with the formal definition of GHE by Armknecht, Katzenbeisser and Peter [8].

**Definition 2** (GHE, Definition 1 in [8])**.** *A public-key encryption scheme $\mathcal{E} = (G, E, D)$ is called* group homomorphic, *if for every* $(\mathsf{pk}, \mathsf{sk}) \leftarrow G(1^\lambda)$*, the plaintext space $\mathcal{M}$ and the ciphertext space $\hat{\mathcal{C}}$ (written in multiplicative notation) are non-trivial groups such that*

- *the set of all encryptions $\mathcal{C} := \{c \in \hat{\mathcal{C}} \mid c \leftarrow E_{\mathsf{pk}}(m), m \in \mathcal{M}\}$ is a non-trivial subgroup of $\hat{\mathcal{C}}$*
- *the restricted decryption $D_{\mathsf{sk}}^* := D_{\mathsf{sk}|\mathcal{C}}$ is a group epimorphism (surjective homomorphism) i.e.,*

$$D_{\mathsf{sk}}^* \text{ is surjective and } \forall c, c' \in \mathcal{C} \; : \; D_{\mathsf{sk}}(c \cdot c') = D_{\mathsf{sk}}(c) \cdot D_{\mathsf{sk}}(c')$$

- $\mathsf{sk}$ *contains an efficient decision function $\delta : \hat{\mathcal{C}} \to \{0, 1\}$ such that*

$$\delta(c) = 1 \iff c \in \mathcal{C}$$

- *the decryption on $\hat{\mathcal{C}} \setminus \mathcal{C}$ returns the symbol $\perp$.*

### 2.4. Identity-Based Group Homomorphic Encryption (IBGHE)

**Definition 3** (Identity-Based Group Homomorphic Encryption (IBGHE), Based on [1])**.** *Let $\mathcal{E} = (G, K, E, D)$ be an IBE scheme with message space $\mathcal{M}$, identity space $\mathcal{I}$ and ciphertext space $\hat{\mathcal{C}}$. The scheme $\mathcal{E}$ is group homomorphic if, for every $(\mathsf{PP}, \mathsf{MSK}) \leftarrow G(1^\lambda)$, every $\mathsf{id} \in \mathcal{I}$, and every $\mathsf{sk}_{\mathsf{id}} \leftarrow K(\mathsf{MSK}, \mathsf{id})$, the message space $(\mathcal{M}, \cdot)$ is a non-trivial group, and there is a binary operation $* : \hat{\mathcal{C}}^2 \to \hat{\mathcal{C}}$ such that the following properties are satisfied for the restricted ciphertext space $\widehat{\mathcal{C}_{\mathsf{id}}} = \{c \in \hat{\mathcal{C}} : D_{\mathsf{sk}_{\mathsf{id}}}(c) \neq \perp\}$:*

**GH.1:**  *The set of all encryptions $\mathcal{C}_{\mathsf{id}} = \{c \mid c \leftarrow E(\mathsf{PP}, \mathsf{id}, m), m \in \mathcal{M}\} \subseteq \widehat{\mathcal{C}_{\mathsf{id}}}$ is a non-trivial group with respect to the operation $*$.*

**GH.2:**  *The restricted decryption $D_{\mathsf{sk}_{\mathsf{id}}}^* := D_{\mathsf{sk}_{\mathsf{id}}|\mathcal{C}_{\mathsf{id}}}$ is surjective and $\forall c, c' \in \mathcal{C}_{\mathsf{id}} \quad D_{\mathsf{sk}_{\mathsf{id}}}(c * c') = D_{\mathsf{sk}_{\mathsf{id}}}(c) \cdot D_{\mathsf{sk}_{\mathsf{id}}}(c')$.*

We are interested in schemes whose plaintext space forms a group and which allow the operation to be homomorphically applied an unbounded number of times. There exist schemes, however, that do not satisfy all the requirements of GHE, namely their ciphertext space does not form a group but instead forms a quasigroup (a group without associativity), such as the Cocks' IBE [10], which was shown to be inherently XOR-homomorphic by Joye [3].

### 2.5. Multi-User Encryption

A multi-user encryption (MUE) scheme is an abstraction from a class of public-key encryption schemes where the public keys of users share common public parameters, whose generation may or may not include a trusted setup, in which case a Trusted Authority (TA) may hold a master decryption key that enables them to decrypt the ciphertexts of any user. An instance of MUE is ElGamal, which does not require a trusted setup or involve a Trusted Authority with a "backdoor", whereas another instance of an MUE is a public-key encryption scheme with a double decryption mechanism (DD-PKE), as defined by Galindo and Herranz [11] where the public parameters are generated along with a master secret key by a TA.

An MUE is a tuple of PPT algorithms $(\mathsf{Setup}, \mathsf{KeyGen}, \mathsf{Enc}, \mathsf{Dec}, \mathsf{mDec})$ with plaintext space $\mathcal{M}$ and ciphertext space $\hat{\mathcal{C}}$, defined as follows:

- $\mathsf{Setup}(1^\lambda)$: takes as input a security parameter $\lambda$ and outputs a pair $(\mathsf{PP}, \mathsf{MSK})$ consisting of public parameters $\mathsf{PP}$ and an optional master secret key $\mathsf{MSK}$, which may be set to $\perp$,

- KeyGen(PP): takes as input the public parameters PP and outputs a pair of public/private keys $(\mathsf{pk}, \mathsf{sk})$.
- Enc(PP, pk, $m$): takes as input the public parameters PP, a user's public key pk and a message $m \in \mathcal{M}$, and outputs a ciphertext $c \in \mathcal{C} \subseteq \widehat{\mathcal{C}}$.
- Dec(PP, sk, $c$): takes as input the public parameters PP, a secret key sk and a ciphertext $c \in \widehat{\mathcal{C}}$, and outputs either a plaintext $m \in \mathcal{M}$ or $\perp$ if decryption fails.
- mDec(PP, MSK, pk, $c$): takes as input the public parameters PP, the master secret key MSK, a user's public key pk and a ciphertext $c \in \widehat{\mathcal{C}}$ and outputs either a plaintext $m \in \mathcal{M}$ or $\perp$ if decryption fails or MSK $= \perp$.

### 2.6. Elliptic Curves over Rings

**Proposition 1** ([5]). *If $N = pq$ is some RSA modulus, i.e., $p$ and $q$ are primes of about the same bit length $\lambda$, then there is an efficient construction of elliptic curves $E : y^2 z = x^3 + axz^2 + bz^3$ over $\mathbb{Z}_{N^2}$ such that $M := \mathsf{lcm}(\#E(\mathbb{Z}_p), \#E(\mathbb{Z}_q))$ has at least two large (of about the same size as $p$ and $q$) prime factors.*

**Lemma 1** ([5]). *As in Proposition 1, let $M \in \mathbb{N}$ have at least two large prime factors (of about $\lambda$ bits). If $\pi(M)$ denotes the product of all small prime factors of $M$, then*

$$\Pr_{s \leftarrow\$ \Pi(M)} \left[ \gcd(s, M) \neq 1 \right] \text{ is negligible in } \lambda$$

*where $\Pi(M) := \{s \in \mathbb{Z}_{N^2} \setminus \{0\} \mid \gcd(s, \pi(M)) = 1\}$.*

### 2.7. Indistinguishability Obfuscation

**Definition 4 (Indistinguishability Obfuscation).** *(Based on Definition 7 from ([12]) A uniform PPT machine $i\mathcal{O}$ is called an indistinguishability obfuscator for every circuit class $\{\mathcal{C}_\kappa\}$ if the following two conditions are met:*

- ***Correctness:*** *For every $\kappa \in \mathbb{N}$, for every $C \in \mathcal{C}_\kappa$, for every $x$ in the domain of $C$, we have that*

$$\Pr\, C'(x) = C(x) : C' \leftarrow i\mathcal{O}(C) = 1.$$

- ***Indistinguishability:*** *For every $\kappa \in \mathbb{N}$, for all pairs of circuits $C_0, C_1 \in \mathcal{C}_\kappa$, if $C_0(x) = C_1(x)$ for all inputs $x$, then for all PPT adversaries $\mathcal{A}$, we have:*

$$|\Pr\, \mathcal{A}(i\mathcal{O}(C_0)) = 1| - |\Pr\, \mathcal{A}(i\mathcal{O}(C_1)) = 1| \leq \mathsf{negl}(\kappa).$$

### 2.8. Puncturable Pseudorandom Function

A puncturable pseudorandom function (PRF) is a constrained PRF (Key, Eval) with an additional PPT algorithm Puncture. Let $n(\cdot)$ and $m(\cdot)$ be polynomials. Our definition here is based on Section 2.5 of [6]. A PRF key $K$ is generated with the PPT algorithm Key, which takes as input the security parameter $\kappa$. The Eval algorithm is deterministic, and on input of a key $K$ and an input string $x \in \{0, 1\}^{n(\kappa)}$, outputs a string $y \in \{0, 1\}^{m(\kappa)}$.

A puncturable PRF allows one to obtain a "punctured" key $K' \leftarrow \mathsf{Puncture}(K, S)$ with respect to a subset of input strings $S \subset \{0, 1\}^{n(\kappa)}$ with $|S| = \mathsf{poly}(\kappa)$. It is required that $\mathsf{Eval}(K, x) = \mathsf{Eval}(K', x) \quad \forall x \in \{0, 1\}^{n(\kappa)} \setminus S$, and for any poly-bounded adversary $(\mathcal{A}_1, \mathcal{A}_2)$ with $S \leftarrow \mathcal{A}_1(1^\kappa) \subset \{0, 1\}^{n(\kappa)}$ and $|S| = \mathsf{poly}(\kappa)$, any key $K \leftarrow \mathsf{Key}(1^\kappa)$, any $K' \leftarrow \mathsf{Puncture}(K, S)$ and any $x \in S$, it holds that

$$\Pr\, \mathcal{A}_2(K', x, \mathsf{Eval}(K, x)) = 1 - \Pr\, \mathcal{A}_2(K', x, u) = 1 \leq \mathsf{negl}(\kappa)$$

where $u \leftarrow\$ \{0, 1\}^{m(\kappa)}$.

### 3. Construction of Anonymous Additively Homomorphic IBE

*3.1. PKTK MUE Scheme*

We now describe the cryptosystem from [5] that is an instance of an MUE and satisfies some interesting properties, including the fact that even the Trusted Authority cannot determine which user a ciphertext is created for (Property 3 [5]), so the scheme is anonymous even to the TA under the hardness of DDH in $E(\mathbb{Z}_{N^2})$. The scheme is very similar to Galbraith's elliptic-curve-based Paillier scheme [13].

- Setup$(1^\lambda)$ : On input of a security parameter $\lambda$, this algorithm generates an RSA modulus $N = pq$ where $p$ and $q$ are primes of about the same bit length $\lambda$. Then it constructs an elliptic curve $E : y^2z = x^3 + axz^2 + bz^3$ over $\mathbb{Z}_{N^2}$ such that $E$ has the properties described in Proposition 1. Furthermore, it chooses a point $Q = (x, y, z) \in E(\mathbb{Z}_{N^2})$ whose order divides $M = \mathsf{lcm}(\#E(\mathbb{Z}_p), \#E(\mathbb{Z}_q))$. It outputs the public parameters PP $:= (N, \pi((M), a, b, Q)$ and the master secret key MSK $:= M$. The plaintext space is $\mathcal{M} = \mathbb{Z}_N$, and the ciphertext space is $\hat{\mathcal{C}} = \langle Q \rangle \times \langle Q, \mathcal{M}_1 \rangle$.

- KeyGen(PP): chooses $s \leftarrow_\$ \mathbb{Z}_M^*$ at random (This can be performed by sampling $s \leftarrow_\$ \Pi(M)$ (which is possible as $\pi(M)$ is included in PP)) and computes $R \leftarrow sQ$. It outputs public key pk $:= R$ and secret key sk $:= s$.

- Enc(PP, pk, $m$): chooses a random value $r \leftarrow_\$ \mathbb{Z}_{N^2}$ and computes the ciphertext $(A, B)$ as
$$A \leftarrow rQ \text{ and } B \leftarrow rR + \mathcal{M}_m.$$

- Dec($pp$, sk, $(A, B)$): outputs
$$m \leftarrow \frac{x(B - sA)}{N}.$$

- mDec(PP, MSK, $(A, B)$) : outputs
$$m \leftarrow \frac{x(MB)}{N} M^{-1} \mod N.$$

*3.2. Our Scheme*

Our scheme is essentially the transformation in [6] applied to the MUE scheme above. We need to define a program $F_{\mathsf{MapPK}}$ that is obfuscated as part of the public parameters. Let $\mathcal{E}$ be an MUE scheme such as the PKTK scheme above, which has message space $\mathbb{Z}_N$. The program $F_{\mathsf{MapPK}}$ takes an identity id and maps it to the public key $\mathsf{pk_{id}}$.

---

**Program $F_{\mathsf{MapPK}}(\mathsf{id})$ :**

1. Compute $r_{\mathsf{id}} \leftarrow \mathsf{PRF.Eval}(K, \mathsf{id})$.
2. Compute $(\mathsf{pk_{id}}, \mathsf{sk_{id}}) \leftarrow \mathcal{E}.\mathsf{KeyGen}(\mathsf{PP}_\mathcal{E}; r_{\mathsf{id}})$.
3. **Output** $\mathsf{pk_{id}}$

---

Let $\mathcal{E}$ be the PKTK MUE scheme. Let $i\mathcal{O}$ be an indistinguishability obfuscator and let PRF be a puncturable PRF. We now define the construction.

- AH.Setup$(1^\lambda)$ : On input of security parameter $\lambda$, compute $(\mathsf{PP}_\mathcal{E}, \mathsf{MSK}_\mathcal{E}) \leftarrow \mathcal{E}.\mathsf{Setup}(1^\lambda)$. Next, generate $K \leftarrow \mathsf{PRF.Gen}(1^\lambda)$ and compute $\mathcal{O} \leftarrow i\mathcal{O}(F_{\mathsf{MapPK}_{\mathsf{PP}_{\mathcal{E},K}}})$. Output $(\mathsf{PP} := (\mathcal{O}, \mathsf{PP}_\mathcal{E}), \mathsf{MSK} := (K, \mathsf{MSK}_\mathcal{E})$.

- AH.KeyGen(MSK, id) : On input of master secret key MSK $:= (K, \mathsf{MSK}_\mathcal{E})$ and an identity id, compute $r_{\mathsf{id}} \leftarrow \mathsf{PRF.Eval}(K, \mathsf{id})$. Next, generate $(\mathsf{pk_{id}}, \mathsf{sk_{id}}) \leftarrow \mathcal{E}.\mathsf{KeyGen}(\mathsf{PP}_\mathcal{E}; r_{\mathsf{id}})$. Output $\mathsf{sk_{id}}$.

- AH.Enc(PP, id, $m$) : On input of public parameters PP, an identity id and a message $m \in \mathbb{Z}_N$, obtain $\mathsf{pk_{id}} \leftarrow \mathcal{O}(\mathsf{id})$ and compute $c \leftarrow \mathcal{E}.\mathsf{Enc}(\mathsf{PP}_\mathcal{E}, \mathsf{pk_{id}}, m)$. Output $c$.

- AH.Dec($\mathsf{sk_{id}}$, $c$): On input of a secret key $\mathsf{sk_{id}}$ for identity id, compute $m \leftarrow \mathcal{E}.\mathsf{Dec}(\mathsf{PP}_\mathcal{E}, \mathsf{sk_{id}}, c)$ and output $m$.

**Theorem 1.** *Assuming indistinguishability obfuscation and the hardness of DDH in $E(\mathbb{Z}_{N^2})$, AH is an anonymous and IND-ID-CPA secure IBE scheme.*

**Proof.** The theorem follows as a consequence of Theorem 1 in [6], where the underlying public-key encryption scheme is replaced with the PKTK MUE scheme whose key privacy and semantic security rely on the hardness of DDH in $E(\mathbb{Z}_{N^2})$. □

This simple construction serves mainly as a possible result for an anonymous homomorphic IBE where the homomorphic operation can be computed without knowing the identity associated with one or more ciphertexts. We leave the construction of more efficient and perhaps even practical schemes of this nature as an open problem.

**4. Anonymous IBE with Rerandomizable Anonymous Encryption Keys**

In this section, we present an anonymous IBE scheme that is a variant of Boneh–Franklin and show that it is both anonymous and IND-ID-RCCA secure. The scheme has two interesting properties: the generation of anonymous keys associated with a particular recipient identity and the rerandomization of such keys. In regard to the former, anonymous keys allow a party to encrypt a message for an unknown recipient; that is, the key hides the identity of the recipient. In regard to the rerandomization of these keys, a rerandomized key is computationally unlinkable to another anonymous key with the same associated identity. Therefore, two anonymous keys for the same identity, where one is obtained by rerandomizing the other, cannot be linked in any way. These properties are essential in our application of anonymous aggregation in the next section. Here, we observe that an essential building block of our construction is an anonymous homomorphic IBE for addition modulo $N$ as realized in the previous section. In fact, an anonymous homomorphic IBE from LWE does not suffice here; a group homomorphic scheme appears to be necessary.

*4.1. Our Construction*

Let $g \in \mathbb{G}$ be a generator of a cyclic group $\mathbb{G}$, and let $g_T \in \mathbb{G}_T$ be a generator of another cyclic group $\mathbb{G}_T$. Both groups are of order $N$, a large semiprime. Now let $e : \mathbb{G} \times \mathbb{G} \to \mathbb{G}_T$ be a non-degenerate bilinear map between $\mathbb{G}$ and $\mathbb{G}_T$ (the target group) such that $g_T = e(g, g)$. The notational convention we follow in this section is to write group elements using uppercase letters whose integer exponent with respect to the generator is the corresponding lowercase letter. Our construction is based around the Boneh–Franklin scheme. We now describe our construction, which serves to illustrate various concepts we would like to establish. Let $H$ be a hash function modeled as a random oracle that maps identity strings to elements of $\mathbb{G}$. The master secret key contains an integer $s \leftarrow_\$ \mathbb{Z}_N$ chosen at setup while the public parameters contain $S \leftarrow g^s$. The other building blocks are an anonymous group homomorphic IBE scheme $\mathcal{E}_m$ that is homomorphic for addition modulo $N$, a NIZK and an IND-CCA2 secure symmetric encryption scheme. Consider a recipient identity id. Then we derive the public key for id as $A \leftarrow H(\mathsf{id}) \in \mathbb{G}$. The encryptor chooses a random integer $r \leftarrow_\$ \mathbb{Z}_N$ and computes $\hat{A} \leftarrow A^r$. Then they compute $\psi_1 \leftarrow \mathcal{E}_m.\mathsf{Enc}(\mathsf{PP}_{\mathsf{IBE}}, \mathsf{id}, r)$ and $z_1 \leftarrow \mathcal{E}_m.\mathsf{Enc}(\mathsf{PP}_{\mathsf{IBE}}, \mathsf{id}, 1_{\mathcal{M}})$. Subsequently, the encryptor chooses a random integer $b \leftarrow_\$ \mathbb{Z}_N$ and computes $B \leftarrow g^b$ and $\psi_2 \leftarrow \mathsf{PKE.Enc}(\mathsf{pk}_T, b; \rho)$ for some randomness $\rho$. Finally, the encryptor generates a NIZK proof $\pi$ that $\psi_2$ encrypts the discrete logarithm of $B$ with respect to base $g$. We derive the symmetric key $k \leftarrow e(\hat{A}^b, S) \in \mathbb{G}_T$ and encrypt the message with the symmetric encryption scheme using the key $k$.

In the real mode, a decryptor with a secret key $\mathsf{sk_{id}} := (S_{\mathsf{id}} := A^s, \mathsf{sk_{IBE,id}} \leftarrow \mathcal{E}_m.\mathsf{KeyGen}(\mathsf{MSK_{IBE}}, \mathsf{id}))$ for identity id, computes $r \leftarrow \mathcal{E}_m.\mathsf{Dec}(\mathsf{sk_{id}}, \psi_1)$ and $k \leftarrow e(B, S_{\mathsf{id}})^r \in \mathbb{G}_T$. In the security proof, when we do not have access to $S_{\mathsf{id}}$, we alternatively derive $k$ as follows. First, we decrypt $\psi_2$ with the trapdoor secret key to obtain $b$ then we compute $k \leftarrow e(\hat{A}^b, S) \in \mathbb{G}_T$.

To generate an anonymous key for an identity, consider the following algorithm:

- $\mathsf{GenAnonKey}(\mathsf{PP}, \mathsf{id})$:

- $r \leftarrow_{\$} \mathbb{Z}_N$
- $\psi \leftarrow \mathcal{E}_m.\mathsf{Enc}(\mathsf{PP}_{\mathsf{IBE}}, \mathsf{id}, r)$
- $z \leftarrow \mathcal{E}_m.\mathsf{Enc}(\mathsf{PP}_{\mathsf{IBE}}, \mathsf{id}, 1_{\mathcal{M}})$
- $A \leftarrow H(\mathsf{id})$
- $\hat{A} \leftarrow A^r$
- Return $\mathsf{AnK} := (\hat{A}, \psi, z)$

An anonymous key AnK lets a party encrypt messages for an unknown intended recipient, which is computationally hidden from the party.

To rerandomize an AnK generated as above, the following algorithm is used:

- $\mathsf{RerandomizeKey}(\mathsf{PP}, \mathsf{AnK})$:
  - Parse AnK as $(\hat{A}, \psi, z)$
  - $r' \leftarrow_{\$} \mathbb{Z}_N$
  - $\hat{A}' \leftarrow \hat{A}^{r'}$
  - $u_1, u_2 \leftarrow_{\$} \mathbb{Z}_N$
  - $\psi' \leftarrow \psi^{r'} * z^{u_1}$
  - $z' \leftarrow z^{u_2}$
  - Return $\mathsf{AnK}' := (\hat{A}', \psi', z')$

The advantage of RerandomizeKey is that given an anonymous key derived with this algorithm from original anonymous key; no party can link the keys and determine that they are related (i.e., have the same intended recipient). The anonymous key is preprended to every ciphertext generated with it, so, therefore, it is advantageous to rerandomize it so ciphertexts are not linked to each other.

We present the scheme formally now. Note that the encryption algorithm may alternatively accept an anonymous key AnK as input instead of a recipient identity.

Algorithm 1 formally describes the scheme.

### 4.2. Security

The scheme cannot be proved IND-ID-CCA2 secure in the conventional sense because the AnK portion of the ciphertext is malleable, and so too is the NIZK proof potentially (unless a non-malleable NIZK is used). We can, however, prove the scheme secure against an adaptive chosen ciphertext attack in a relaxed model, namely the notion IND-ID-RCCA .

**Theorem 2.** *Assuming $\mathcal{E}_m$ is IND-ID-CPA secure,* PKE *is IND-CPA secure and* NIZK *is a sound and zero-knowledge NIZK, then our scheme is IND-ID-RCCA secure under the hardness of DBDH in the random oracle model.*

**Algorithm 1** Our IBE scheme with rerandomizable anonymous keys.

Setup $(1^\lambda)$
  $(\mathsf{PP_{IBE}}, \mathsf{MSK_{IBE}}) \leftarrow \mathcal{E}_m.\mathsf{Setup}(1^\lambda)$
  $(\mathsf{pk}_T, \mathsf{sk}_T) \leftarrow \mathsf{PKE.Gen}(1^\lambda)$
  $H \leftarrow_\$ \mathcal{H}$
  $s \leftarrow_\$ \mathbb{Z}_N$
  $S \leftarrow g^s$
  $\mathsf{CRS} \leftarrow \mathsf{NIZK.CRSGen}(1^\lambda)$
  Return $(\mathsf{PP} := (H, S, \mathsf{PP_{IBE}}, \mathsf{pk}_T, \mathsf{CRS}), \mathsf{MSK} := (K, s, \mathsf{MSK_{IBE}}, \mathsf{sk}_T))$

KeyGen$(\mathsf{MSK}, \mathsf{id})$
  $A \leftarrow H(\mathsf{id})$
  $S_{\mathsf{id}} \leftarrow A^s$
  $\mathsf{sk_{IBE,id}} \leftarrow \mathcal{E}_m.\mathsf{KeyGen}(\mathsf{MSK_{IBE}}, \mathsf{id})$
  Return $\mathsf{sk_{id}} := (S_{\mathsf{id}}, \mathsf{sk_{IBE,id}})$

Enc$(\mathsf{PP}, \mathsf{id}, m)$
  $r \leftarrow_\$ \mathbb{Z}_N$
  $\psi_1 \leftarrow \mathcal{E}_m.\mathsf{Enc}(\mathsf{PP_{IBE}}, \mathsf{id}, r)$
  $z \leftarrow \mathcal{E}_m.\mathsf{Enc}(\mathsf{PP_{IBE}}, \mathsf{id}, 1_\mathcal{M})$
  $A \leftarrow H(\mathsf{id})$
  $\hat{A} \leftarrow A^r$
  $b \leftarrow_\$ \mathbb{Z}_N$
  $B \leftarrow g^b$
  $\rho \leftarrow_\$ \{0,1\}^{\ell_\rho}$ // where $\ell_\rho$ is the
    length of randomness required for PKE.Enc
  $\psi_2 \leftarrow \mathsf{PKE.Enc}(\mathsf{pk}_T, b; \rho)$
  $\pi \leftarrow \mathsf{NIZK.Prove}(\mathsf{CRS}, (g, B, \mathsf{pk}_T, \psi_2), (b, \rho))$
    // the NIZK uses relation $R$ (below)
  $k \leftarrow e(\hat{A}^b, S)$
  $\psi_3 \leftarrow \mathsf{SKE.Enc}(k, \psi_1 \parallel m)$
  Return $c := (\hat{A}, \psi_1, z, B, \psi_2, \pi, \psi_3)$

Dec$(\mathsf{sk_{id}}, c)$
  $(S_{\mathsf{id}}, \mathsf{sk_{IBE,id}}) \leftarrow \mathsf{sk_{id}}$
  $(\hat{A}, \psi_1, z, B, \psi_2, \pi, \psi_3) \leftarrow c$
  If $\mathsf{NIZK.Verify}(\mathsf{CRS}, (g, B, \mathsf{pk}_T, \psi_2), \pi) \neq 1$
    Return $\bot$
  $r \leftarrow \mathcal{E}_m.\mathsf{Dec}(\mathsf{sk_{IBE,id}}, \psi_1)$
  $If\, \hat{A} \neq A^r$
    Return $\bot$
  $k \leftarrow e(S_{\mathsf{id}}, B)^r$
  Return $\mathsf{SKE.Dec}(k, \psi_3)$

**Relation** $R(\mathsf{stmt} := (g, B, \mathsf{pk}_T, \psi_2), w := (b, \rho))$
  Return $B = g^b \wedge \psi_2 = \mathsf{PKE.Enc}(\mathsf{pk}_T, b; \rho)$

**Proof.** We prove the theorem by means of a hybrid argument. We start with a real system that encrypts the first challenge message $m_0$, and move to a hybrid that encrypts the second challenge message $m_1$.

**Hybrid 0**: This is the real system that encrypts the challenge message $m_0$. Let $k$ be the symmetric key used to produce the symmetric ciphertext $\psi_3$.

**Hybrid 1**: The change we make in this hybrid is to how $\psi_1$ is generated. Instead of encrypting randomness $r$, we choose another uniform random element $s$ and produce $\psi_1$

as an IBE encryption of $s$. We still use the previous symmetric key $k$ to produce $\psi_3$, which is a symmetric encryption of $\psi_1 \parallel m_0$.

The indistinguishability between Hybrids 0 and 1 follows from the semantic security of the $\mathcal{E}_m$. In the reduction, we use the "trapdoor" mode discussed earlier to derive the symmetric key; that is, for a typical ciphertext, we decrypt $\psi_!$ to obtain $b$ and compute $e(\hat{A}, S)^b$. When we decrypt $\psi_3$, we check if the first component of the plaintext matches $\psi_1$; otherwise, we output $\perp$. Secondly, if the second component is $m_0$ or $m_1$, we output "test" as is required in IND-ID-RCCA. If the ciphertext we gave the adversary is queried for decryption, then we also output "test".

**Hybrid 2**: The change we make in this hybrid is to how $\psi_1$ is generated. We compute it instead as an encryption of some uniformly random element $z \neq b$ but still use $k$ (as in the previous hybrid) to produce $\psi_3$.

Hybrids 1 and 2 are indistinguishable from the IND-CCA2 security of PKE. In the reduction, we return the original approach (i.e., the "real" mode) to compute the symmetric key.

**Hybrid 3**: The change we make in this hybrid is to generate the symmetric key uniformly at random.

The indistinguishability of Hybrids 2 and 3 follows from the hardness of DBDH.

**Hybrid 4**: In this hybrid, we change how $\psi_3$ is produced. Instead of encrypting $\psi_1 \parallel m_0$, we encrypt $\psi_1 \parallel m_1$.

The indistinguishability of Hybrids 3 and 4 follows from the iND-CCA2 security of the symmetric encryption scheme. We are now in a hybrid where the second challenge message $m_1$ is encrypted. The remaining hybrids reverse the changes in Hybrids 1–3 until we arrive at a hybrid that is the real system that encrypts the challenge message $m_1$. This completes our proof. □

**Corollary 1.** *Assuming $\mathcal{E}_m$ is an IND-ID-CPA secure anonymous IBE, then our scheme is anonymous.*

This is an immediate consequence of the semantic security and anonymity of $\mathcal{E}_m$.

## 5. Identity-Based Anonymous Aggregation

In an identity-based anonymous aggregation protocol, a collection of nodes encrypt data for different recipients and forward them to their neighbors. The intended recipient, along with an aggregator, is able to extract the following grouping, functional unit or "package", comprising the tuple $(h, v, z)$, which we define momentarily. Let $\mathcal{E}$ be an anonymous IBGHE scheme (such as AH in Section 3), and let $H$ be a collision-resistant function. Furthermore, let id be the recipient's identity. Then we have $h = H(\mathrm{id})$, $v \leftarrow \mathcal{E}.\mathrm{Enc}(\mathrm{PP}_{\mathcal{E}}, \mathrm{id}, m)$ and $z \leftarrow \mathcal{E}.\mathrm{Enc}(\mathrm{PP}_{\mathcal{E}}, \mathrm{id}, \mathsf{o})$. For two such tuples, $c := (h, v, z)$ and $c' := (h', v', z')$, the aggregation algorithm is defined in Algorithm 2.

---
**Algorithm 2** Aggregation algorithm in P-type setting.

$\mathrm{Agg.Aggregate}(c, c')$.
  $(h, v, z) \leftarrow c$
  $(h', v', z') \leftarrow c'$
  If $h \neq h'$:
    Output $\perp$
  $s_1, s_2 \leftarrow_\$ \mathbb{Z}_N$
  $v'' \leftarrow v * v' * z^{s_1}$
  $z'' \leftarrow z^{s_2}$
  Return $c'' := (h'' := h, v'', z'')$

---

The hash of the recipient's identity $h$ allows an aggregator to determine whether two ciphertexts have the same intended recipient, in which case, the hash components are equal,

and aggregation can be performed; otherwise, aggregating both ciphertexts would produce an invalid result. With this approach, we obtain one-way anonymity. The $v$ component is an $\mathcal{E}$ encryption under the recipient's identity of the plaintext value. For the sake of simplicity, we are assuming the plaintext space is $\mathcal{M} := \mathbb{Z}_N$. For referential convenience, we designate this type of scheme P-type.

Now, an alternative approach is to exclude the hash component from this tuple such that an aggregator cannot learn anything about the recipient's identity, nor can it determine whether two ciphertexts have the same recipient. As such, aggregation is always performed, but we need some way for the decryptor to establish whether a ciphertext is valid or has been likely contaminated through aggregation with a different identity. A solution to this emerges when the plaintext space is exponentially large, as is the case here. The idea is to include additional encryption $\bar{v}$ of $-m$ where the underlying plaintext of $v$ is $m$ such that $v * \bar{v}$ decrypts to zero (or $1_{\mathcal{M}}$, the identity element). The decryptor discards a ciphertext as invalid if $v * \bar{v}$ does not decrypt to zero. Homomorphically adding (pairwise) a pair of ciphertexts $(v', \bar{v}')$ associated with another identity results in a pair of encryptions of random values in $\mathbb{Z}_N$. Therefore, the resulting ciphertext will be rejected as invalid by the decryptor with overwhelming probability. For referential convenience, we designate this type of scheme F-type. The aggregation algorithm for this type is shown in Algorithm 3.

---

**Algorithm 3** Aggregation algorithm in F-type setting.

Agg.Aggregate$(c, c')$.
 $(v, \bar{v}, z) \leftarrow c$
 $(v', \bar{v}', z') \leftarrow c'$
 $s_1, s_2, s_3 \leftarrow_\$ \mathbb{Z}_N$
 $v'' \leftarrow v * v' * z^{s_1}$
 $\bar{v}'' \leftarrow \bar{v} * \bar{v}' * z^{s_2}$
 $z'' \leftarrow z^{s_3}$
 Return $c'' := (v'', \bar{v}'', z'')$

---

Since any party who obtains the ciphertext tuple as above can modify the underlying plaintext (malleability), we may wish to restrict this ability to a subset of authorized parties, which we refer to as *aggregators*. While a suitable means of access control for granting such authorization to aggregators is beyond the scope of this work (e.g., ABE and related primitives may be of import), we describe a simplified paradigm that can be adapted and extended as required. Typically, we would expect the ciphertext tuple above to be encrypted with a non-malleable encryption scheme, such as an IND-CCA2 secure symmetric-key encryption scheme denoted by SKE. Moreover, a random symmetric key $\kappa$ is first generated, and the tuple $c$ is then encrypted, i.e., we have $\psi \leftarrow \mathsf{SKE.Enc}(\kappa, c)$. The natural question is, then, how does one obtain $\kappa$? Note that both authorized aggregators and the recipient must be able to access $\kappa$. First, an appropriate means of access control can be employed to allow authorized aggregators to access $\kappa$, a subject that, as aforementioned, is outside the scope of this work. Secondly, and most importantly, the intended recipient must be able to access $\kappa$. The challenge arises for intermediate aggregators who need to encrypt a fresh $\kappa$ under the recipient's identity, which is hidden from them due to the desired property of anonymity. It is apparent from the proof of *aggregation validity* that the IBE scheme in which $\kappa$ is encrypted must be secure against adaptive chosen ciphertext attacks. Aggregation validity is a property that is defined in the next section and informally means that no efficient adversary who is given an encryption of a message $m$ and who is neither an authorized aggregator nor the intended recipient can produce a valid ciphertext that encrypts a targeted modification of $m$, that is, $t \cdot m$ for some a priori decided $t \neq 1_{\mathcal{M}}$.

We now formalize the identity-based anonymous aggregation (IBAA) in a simplified form where the authorization of aggregators is based on symmetric encryption, which is sufficient for our purposes, but we note this may be replaced with a more complex form of access control accommodated by a more generalized definition.

**Definition 5.** *An identity-based anonymous aggregation (IBAA) protocol $\mathcal{P}$ consists of the following PPT algorithms:*

- $\mathsf{Setup}(1^\lambda)$*: On input of a security parameter $\lambda$, generate public parameters $\mathsf{PP}$ and master secret key $\mathsf{MSK}$. Output $(\mathsf{PP}, \mathsf{MSK})$.*
- $\mathsf{KeyGen}(\mathsf{MSK}, \mathsf{id})$*: On input of a master secret key $\mathsf{MSK}$ and an identity $\mathsf{id}$, output a secret key $\mathsf{sk}_{\mathsf{id}}$ for identity $\mathsf{id}$.*
- $\mathsf{Authorize}(\mathsf{sk}_{\widetilde{\mathsf{id}}})$*: On input of a secret key $\mathsf{sk}_{\widetilde{\mathsf{id}}}$ for identity $\widetilde{\mathsf{id}}$, output an authorization key that permits aggregation on ciphertexts generated by a source (sender) with identity $\widetilde{\mathsf{id}}$.*
- $\mathsf{Enc}(\mathsf{PP}, \mathsf{sk}_{\widetilde{\mathsf{id}}}, \mathsf{id}, m)$*: On input of public parameters $\mathsf{PP}$, a secret key for the source (sender) $\mathsf{sk}_{\widetilde{\mathsf{id}}}$ whose identity is $\widetilde{\mathsf{id}}$, a recipient identity $\mathsf{id}$ and message $m \in \mathcal{M}$, produce a ciphertext $c$ that encrypts $m$ under identity $\mathsf{id}$ and output $c$.*
- $\mathsf{Dec}(\mathsf{sk}_{\mathsf{id}}, c)$*: On input of a secret key $\mathsf{sk}_{\mathsf{id}}$ for identity $\mathsf{id}$ and a ciphertext $c$, output a message $m \in \mathcal{M}$ if $c$ is a valid ciphertext for identity $\mathsf{id}$; otherwise, output $\bot$.*
- $\mathsf{Aggregate}(\mathsf{PP}, \mathsf{sk}_{\widetilde{\mathsf{id}}}, (\mathsf{ak}_1, c_1), (\mathsf{ak}_2, c_2))$*: On input of public parameters $\mathsf{PP}$, the aggregator's secret key $\mathsf{sk}_{\widetilde{\mathsf{id}}}$ for their identity $\widetilde{\mathsf{id}}$ and two ciphertexts $c_1$ and $c_2$ with corresponding authorization keys $\mathsf{ak}_1$ and $\mathsf{ak}_2$ (it may be the case that $\mathsf{ak}_1 = \mathsf{ak}_2$) that permit aggregation, if $\mathsf{ak}_1$ permits aggregation on $c_1$ and $\mathsf{ak}_2$ permits aggregation on $c_2$, then output $c'$ such that $\mathsf{Dec}(\mathsf{sk}_{\mathsf{id}}, c') = \mathsf{Dec}(\mathsf{sk}_{\mathsf{id}}, c_1) * \mathsf{Dec}(\mathsf{sk}_{\mathsf{id}}, c_2)$ for some operation $*$ (typically for an abelian group). Otherwise, output $\bot$. Additionally, in order to perform aggregation on $c'$, a party needs an authorization key from $\widetilde{\mathsf{id}}$.*

This primitive is very similar to homomorphic IBE, except there are a few notable differences. First, only senders who are authorized by the TA can encrypt messages, which can be decrypted by the recipient if they have received a secret key from the TA for their identity. Secondly, aggregation is possible on a sender's ciphertext only if the aggregator has received an authorization key from the sender.

**Correctness:** *For $i \in \{1, 2\}$, all $(\mathsf{PP}, \mathsf{MSK}) \leftarrow \mathsf{Setup}(1^\lambda)$, all identities $\mathsf{id}_i^* \in \mathcal{I}$ (senders), $\widetilde{\mathsf{id}} \in \mathcal{I}$ (aggregator) and $\mathsf{id} \in \mathcal{I}$ (recipient), all $\mathsf{sk}_{\mathsf{id}_i^*} \leftarrow \mathsf{KeyGen}(\mathsf{MSK}, \mathsf{id}_i^*)$, all $\mathsf{sk}_{\widetilde{\mathsf{id}}} \leftarrow \mathsf{KeyGen}(\mathsf{MSK}, \widetilde{\mathsf{id}})$, all $\mathsf{sk}_{\mathsf{id}} \leftarrow \mathsf{KeyGen}(\mathsf{MSK}, \mathsf{id})$, all $m_i \in \mathcal{M}$, all $c_i \leftarrow \mathsf{Enc}(\mathsf{PP}, \mathsf{id}_i^*, \mathsf{id}, m_i)$ and any $\mathsf{ak}_i$, then*

$$\mathsf{Dec}(\mathsf{sk}_{\mathsf{id}}, \mathsf{Aggregate}(\mathsf{PP}, \mathsf{sk}_{\widetilde{\mathsf{id}}}, (\mathsf{ak}_1, c_1), (\mathsf{ak}_2, c_2))) = m_1 * m_2$$

*iff $\mathsf{ak}_i \in \mathsf{Authorize}(\mathsf{sk}_{\mathsf{id}_i^*})$ (except with negligible probability) where $\mathcal{I}$ is the identity space. More precisely, the second part of the iff in the above condition is actually a security condition, which we now treat on its own.*

**Definition 6.** *An IBAA scheme is said to satisfy (selective) aggregation validity if for all $t \neq \mathsf{o} \in \mathcal{M}$, the advantage of any PPT adversary $\mathcal{A} = (\mathcal{A}_1, \mathcal{A}_2)$ is negligible in the security parameter where the advantage is defined as follows:*

$$\mathsf{Adv}_{\mathcal{A}, \mathsf{AV}} = \Pr\Big[ \mathsf{Dec}(\mathsf{sk}_{\mathsf{id}}, c') \to t * m : \begin{array}{l} (\mathsf{PP}, \mathsf{MSK}) \leftarrow \mathsf{Setup}(1^\lambda), \\ (\widetilde{\mathsf{id}}, \mathsf{id}) \leftarrow \mathcal{A}_1(1^\lambda), \\ m \leftarrow_\$ \mathcal{M}, \\ \mathsf{sk}_{\widetilde{\mathsf{id}}} \leftarrow \mathsf{KeyGen}(\mathsf{MSK}, \widetilde{\mathsf{id}}), \\ \mathsf{sk}_{\mathsf{id}} \leftarrow \mathsf{KeyGen}(\mathsf{MSK}, \mathsf{id}), \\ c \leftarrow \mathsf{Enc}(\mathsf{PP}, \mathsf{sk}_{\widetilde{\mathsf{id}}}, \mathsf{id}, m), \\ c' \leftarrow \mathcal{A}_2^{\mathcal{O}}(\mathsf{PP}, c) \end{array} \Big]$$

*where $\mathcal{O} = \mathsf{KeyGen}(\mathsf{MSK}, \cdot)$ except queries cannot be made for identities $\widetilde{\mathsf{id}}$ and $\mathsf{id}$. It is assumed that $|\mathcal{M}|$ is exponentially large and the min-entropy of $\mathcal{M}$ is sufficiently higher than the security parameter.*

**Definition 7.** *An IBAA scheme is said to satisfy (selective) strong unlinkability if the advantage of any PPT adversary* $\mathcal{A} = (\mathcal{A}_1, \mathcal{A}_2)$ *is negligible in the security parameter where the advantage is defined as follows:*

$$
\begin{aligned}
\mathsf{Adv}_{\mathcal{A},\mathsf{UL}} = \quad & \Pr \mathcal{A}_2^{\mathcal{O}}(\mathsf{PP}, c', c'', c'') \to 1 : \quad (\mathsf{PP}, \mathsf{MSK}) \leftarrow \mathsf{Setup}(1^\lambda), \\
& (\widetilde{\mathsf{id}}, \widetilde{\mathsf{id}'}, \widetilde{\mathsf{id}''}, m, m', \mathsf{id}) \leftarrow \mathcal{A}_1(1^\lambda), \\
& \mathsf{sk}_{\widetilde{\mathsf{id}}} \leftarrow \mathsf{KeyGen}(\mathsf{MSK}, \widetilde{\mathsf{id}}), \\
& \mathsf{sk}_{\widetilde{\mathsf{id}'}} \leftarrow \mathsf{KeyGen}(\mathsf{MSK}, \widetilde{\mathsf{id}'}), \\
& \mathsf{sk}_{\widetilde{\mathsf{id}''}} \leftarrow \mathsf{KeyGen}(\mathsf{MSK}, \widetilde{\mathsf{id}''}), \\
& \mathsf{ak} \leftarrow \mathsf{Authorize}(\mathsf{sk}_{\widetilde{\mathsf{id}}}), \\
& \mathsf{ak}' \leftarrow \mathsf{Authorize}(\mathsf{sk}_{\widetilde{\mathsf{id}'}}), \\
& c \leftarrow \mathsf{Enc}(\mathsf{PP}, \mathsf{sk}_{\widetilde{\mathsf{id}}}, \mathsf{id}, m), \\
& c' \leftarrow \mathsf{Enc}(\mathsf{PP}, \mathsf{sk}_{\widetilde{\mathsf{id}'}}, \mathsf{id}, m'), \\
& c'' \leftarrow \mathsf{Aggregate}(\mathsf{PP}, \mathsf{sk}_{\widetilde{\mathsf{id}''}}, (\mathsf{ak}, c), (\mathsf{ak}', c'))
\end{aligned}
$$

$$
\begin{aligned}
- \Pr \mathcal{A}_2^{\mathcal{O}}(\mathsf{PP}, c', c'', c'') \to 1 : \quad & (\mathsf{PP}, \mathsf{MSK}) \leftarrow \mathsf{Setup}(1^\lambda), \\
& (\widetilde{\mathsf{id}}, \widetilde{\mathsf{id}'}, \widetilde{\mathsf{id}''}, m, m', \mathsf{id}) \leftarrow \mathcal{A}_1(1^\lambda), \\
& \mathsf{sk}_{\widetilde{\mathsf{id}}} \leftarrow \mathsf{KeyGen}(\mathsf{MSK}, \widetilde{\mathsf{id}}), \\
& \mathsf{sk}_{\widetilde{\mathsf{id}'}} \leftarrow \mathsf{KeyGen}(\mathsf{MSK}, \widetilde{\mathsf{id}'}), \\
& \mathsf{sk}_{\widetilde{\mathsf{id}''}} \leftarrow \mathsf{KeyGen}(\mathsf{MSK}, \widetilde{\mathsf{id}''}), \\
& \mathsf{ak} \leftarrow \mathsf{Authorize}(\mathsf{sk}_{\widetilde{\mathsf{id}}}), \\
& \mathsf{ak}' \leftarrow \mathsf{Authorize}(\mathsf{sk}_{\widetilde{\mathsf{id}'}}), \\
& c \leftarrow \mathsf{Enc}(\mathsf{PP}, \mathsf{sk}_{\widetilde{\mathsf{id}}}, \mathsf{id}, m), \\
& c' \leftarrow \mathsf{Enc}(\mathsf{PP}, \mathsf{sk}_{\widetilde{\mathsf{id}'}}, \mathsf{id}, m'), \\
& c'' \leftarrow \mathsf{Enc}(\mathsf{PP}, \mathsf{sk}_{\widetilde{\mathsf{id}''}}, \mathsf{id}, m * m')
\end{aligned}
$$

*where* $\mathcal{O} = \mathsf{KeyGen}(\mathsf{MSK}, \cdot)$; *note that queries can be made for identity* id.

**Definition 8.** *An IBAA scheme is said to be one-way anonymous if the advantage of any PPT adversary* $\mathcal{A} = (\mathcal{A}_1, \mathcal{A}_2)$ *is negligible in the security parameter where the advantage is defined as follows:*

$$
\begin{aligned}
\mathsf{Adv}_{\mathcal{A},\mathsf{OW\text{-}ANON}} = \quad & \Pr \mathcal{A}_2^{\mathcal{O}}(\mathsf{PP}, c) \to \mathsf{id} : \quad (\mathsf{PP}, \mathsf{MSK}) \leftarrow \mathsf{Setup}(1^\lambda), \\
& (\widetilde{\mathsf{id}}, m) \leftarrow \mathcal{A}_1(1^\lambda), \\
& \mathsf{id} \leftarrow^\$ \mathcal{I}, \\
& \mathsf{sk}_{\widetilde{\mathsf{id}}} \leftarrow \mathsf{KeyGen}(\mathsf{MSK}, \widetilde{\mathsf{id}}), \\
& c \leftarrow \mathsf{Enc}(\mathsf{PP}, \mathsf{sk}_{\widetilde{\mathsf{id}}}, \mathsf{id}, m)
\end{aligned}
$$

*where* $\mathcal{O} = \mathsf{KeyGen}(\mathsf{MSK}, \cdot)$. *It is assumed that* $\mathcal{I}$ *is exponentially large and the min-entropy of* $\mathcal{I}$ *is sufficiently higher than the security parameter.*

## 6. Construction of IBAA

We now present a construction of the primitive defined in Section 5. Our construction requires an anonymous homomorphic IBE scheme $\mathcal{E}_m$ for the plaintext values, a collision-resistant hash function family, a symmetric encryption scheme $\mathcal{E}_{\mathsf{SKE}}$, a PRF and an anonymous IBE $\mathcal{E}_k$ for encrypting the keys. Let $\mathcal{H}$ be a family of collision-resistant hash functions. Our IBAA scheme is shown in Algorithm 4.

---

**Algorithm 4** Our IBAA scheme—first five algorithms.

---

$\mathsf{Agg.Setup}(1^\lambda)$.
 $K \leftarrow \mathsf{PRF.Gen}(1^\lambda)$
 $(\mathsf{PP_{IBE}}, \mathsf{MSK_{IBE}}) \leftarrow \mathcal{E}_m.\mathsf{Setup}(1^\lambda)$
 $(\mathsf{PP'_{IBE}}, \mathsf{MSK'_{IBE}}) \leftarrow \mathcal{E}_k.\mathsf{Setup}(1^\lambda)$
 $H \leftarrow_\$ \mathcal{H}$
 Return $(\mathsf{PP} := (H, \mathsf{PP_{IBE}}, \mathsf{PP'_{IBE}}), \mathsf{MSK} := (K, \mathsf{MSK_{IBE}}, \mathsf{MSK'_{IBE}}))$

$\mathsf{Agg.KeyGen}(\mathsf{MSK}, \mathsf{id})$.
 $r_\alpha \leftarrow \mathsf{PRF.Eval}(K, \mathsf{id} \parallel \text{'A'})$
 $\alpha_{\mathsf{id}} \leftarrow \mathcal{E}_{\mathsf{SKE}}.\mathsf{Gen}(1^\lambda; r_\alpha)$
 $\mathsf{sk_{IBE}} \leftarrow \mathcal{E}_m.\mathsf{KeyGen}(\mathsf{MSK_{IBE}}, \mathsf{id})$
 $\mathsf{sk'_{IBE}} \leftarrow \mathcal{E}_k.\mathsf{KeyGen}(\mathsf{MSK'_{IBE}}, \mathsf{id})$
 Return $\mathsf{sk_{id}} := (\alpha_{\mathsf{id}}, \mathsf{sk_{IBE}}, \mathsf{sk'_{IBE}})$

$\mathsf{Agg.Authorize}(\mathsf{sk}_{\widetilde{\mathsf{id}}})$.
 $(\alpha_{\widetilde{\mathsf{id}}}, \mathsf{sk_{IBE}}, \mathsf{sk'_{IBE}}) \leftarrow \mathsf{sk}_{\widetilde{\mathsf{id}}}$
 Return $\mathsf{ak}_{\widetilde{\mathsf{id}}} := \alpha_{\widetilde{\mathsf{id}}}$

$\mathsf{Agg.Enc}(\mathsf{PP}, \mathsf{sk}_{\widetilde{\mathsf{id}}}, \mathsf{id}, m)$.
 $(\alpha_{\widetilde{\mathsf{id}}}, \mathsf{sk_{IBE}}, \mathsf{sk'_{IBE}}) \leftarrow \mathsf{sk}_{\widetilde{\mathsf{id}}}$
 $\kappa \leftarrow \mathcal{E}_{\mathsf{SKE}}.\mathsf{Gen}(1^\lambda)$
 $h \leftarrow H(\mathsf{id})$
 $c_1 \leftarrow \mathcal{E}_{\mathsf{SKE}}.\mathsf{Enc}(\alpha_{\widetilde{\mathsf{id}}}, \kappa)$
 $c_2 \leftarrow \mathcal{E}_k.\mathsf{Enc}(\mathsf{PP'_{IBE}}, \mathsf{id}, \kappa)$
 $v \leftarrow \mathcal{E}_m.\mathsf{Enc}(\mathsf{PP_{IBE}}, \mathsf{id}, m)$
 $z \leftarrow \mathcal{E}_m.\mathsf{Enc}(\mathsf{PP_{IBE}}, \mathsf{id}, 1_{\mathcal{M}})$
 $c_3 \leftarrow \mathcal{E}_{\mathsf{SKE}}.\mathsf{Enc}(\kappa, (h, v, z))$
 Return $c := (c_1, c_2, c_3)$

$\mathsf{Agg.Dec}(\mathsf{sk_{id}}, c)$.
 $(\alpha_{\mathsf{id}}, \mathsf{sk_{IBE}}, \mathsf{sk'_{IBE}}) \leftarrow \mathsf{sk_{id}}$
 $\kappa \leftarrow \mathcal{E}_k.\mathsf{Dec}(\mathsf{sk'_{IBE}}, c_2)$
 $t \leftarrow \mathcal{E}_{\mathsf{SKE}}.\mathsf{Dec}(\kappa, c_3)$
 If $t = \bot$:
  Return $\bot$
 $(h, v, z) \leftarrow t$
 $m \leftarrow \mathcal{E}_m.\mathsf{Dec}(\mathsf{sk_{IBE}}, v)$
 Return $m$

---

We now prove an important result.

**Theorem 3.** *Assuming $\mathcal{E}_k$ is IND-ID-RCCA secure and* SKE *is IND-CCA2 secure, then the IBAA scheme in Algorithm 4 satisfies aggregation validity.*

**Proof.** We prove the theorem via a hybrid argument. To avoid repetition and to make the analysis more concise, we describe some notations for things that are common to all steps in the argument. For each step, we need to construct a simulator that uses an adversary $\mathcal{A}$ against selective aggregation validity in either the hybrid from the step in question or the previous hybrid to attack the security of one of the underlying primitives. However, the security games for each of these primitives involve an adversary outputting a guess bit, whereas adversary $\mathcal{A}$ outputs a ciphertext $c'$. Therefore, an essential part of the reduction is to show how we convert this ciphertext $c'$ into a bit $b' \in \{0, 1\}$ such that either $b'$ or its complement can be sent to the challenger to break the security of the underlying primitive.

For the sake of brevity in the reductions below, we simply describe how $b'$ is computed from $c'$.

**Hybrid 0**: This is the real system.

**Hybrid 1**: In this hybrid, we change $c_1$ to the encryption of a uniformly random and independent element.

Indistinguishability follows from the IND-CCA2 security of the symmetric encryption scheme. The reduction, in this case, is straightforward.

**Hybrid 2**: In this hybrid, we change the $c_2$ component of the ciphertext to an encryption of a random element drawn from the message space of the $\mathcal{E}_k$ scheme. Therefore, instead of encrypting $\kappa$, we encrypt a random element $\rho$.

We can use an adversary that has a non-negligible advantage distinguishing between Hybrids 0 and 1 to construct an adversary that has a non-negligible advantage against the IND-ID-RCCA security of $\mathcal{E}_k$. The reduction is as follows. First, we run $\mathcal{A}_1$ to obtain $(\widetilde{id}, id)$. We sample $m \leftarrow_\$ \mathcal{M}$. We run Setup and all steps of the encryption algorithm except the step that generates $c_2$. Therefore, we, for example, generate $\kappa$, $c_1$ and $c_3$. We set $\mu_0 \leftarrow \kappa$ and $\mu_1 \leftarrow \rho$ where $\rho$ is a uniform random element in the message space of $\mathcal{E}_k$ and send the pair of messages $(\mu_0, \mu_1)$ to the IND-ID-RCCA challenger. We receive a challenge ciphertext $e$, and we set $c_1 \leftarrow e$ and set $c \leftarrow (c_1, c_2, c_3)$. Then we run $\mathcal{A}_2$ with the public parameters and ciphertext $c$ and obtain $c'$. We parse $c'$ as $(c_1', c_2', c_3')$. Then the reduction sends $c_1'$ to the IND-ID-RCCA decryption oracle, and if the oracle responds with test, then check if $c_3'$ is decryptable with $\kappa$ or $\rho$ and let $\mu$ be the tuple obtained, or else if the oracle responds with a plaintext $k$, check if $c_3'$ is decryptable with $k$ and set $\mu$ to be the tuple returned. Otherwise, set $\mu \leftarrow \perp$. Finally, the guess bit $b'$ is computed as $b' \leftarrow \mu \neq \perp \wedge .\mathcal{E}_m.\mathsf{Dec}(\mathsf{sk}_{\mathsf{IBE}}, \mu.v) = m * t$, where $\mathsf{sk}_{\mathsf{IBE}}$ is the key we have derived in the simulation. Indistinguishability follows from the IND-ID-RCCA security of $\mathcal{E}_k$.

**Hybrid 3**: In this hybrid, we change the $c_3$ component of the ciphertext to an encryption of a random element drawn from the message space of the SKE scheme.

In the reduction, we parse $c'$ as $(c_1', c_2', c'3)$ and decrypt $c_2'$ with the secret key derived in the simulation to obtain $\kappa$. If $\kappa$ decrypts $c_3'$, set $\mu$ to the resulting tuple. Otherwise, send $c_3'$ to the IND-CCA2 decryption oracle and set $\mu$ to the response. Finally, the guess bit $b'$ is computed as $b' \leftarrow \mu \neq \perp \wedge .\mathcal{E}_m.\mathsf{Dec}(\mathsf{sk}_{\mathsf{IBE}}, \mu.v) = m * t$ where $\mathsf{sk}_{\mathsf{IBE}}$ is the key we have derived in the simulation. Indistinguishability follows from the IND-CCA2 security of the SKE scheme.

The adversary has a negligible advantage in this game since the ciphertext $c$ does not contain any information about $m$. The result follows. $\square$

We have omitted the aggregation algorithm from Algorithm 4 since this varies depending on whether we target the P-type or F-type setting. Our goal is to achieve strong unlinkability, aggregation validity and (one-way/full) anonymity in the (P-type/F-type) settings.

*6.1. P-Type Setting*

We can, however, readily obtain strong unlinkability together with aggregation validity in the P-type setting of one-way anonymity, which we will now describe. Unfortunately, our approach is inherently restricted to one-way anonymity, leaving open the problem of achieving strong unlinkability and aggregation validity in the F-type setting of full anonymity; we will tackle this problem later. Our approach for the P-type setting involves instantiating $\mathcal{E}_k$ with an IND-ID-CCA2 secure IBE scheme. The hash of the target identity $h$ in the tuple encrypted by $c_3$ is used as an identity string; that is, $c_2$ is an encryption with $\mathcal{E}_k$ under identity string $h$ of the symmetric key $\kappa$. The ciphertext component $c_3$ is an encryption of the tuple $(h, v, z)$. The aggregation algorithm for our IBAA scheme in this setting is given in Algorithm 5.

---

**Algorithm 5** Our IBAA scheme aggregation algorithm for P-type setting.

---

$\mathsf{Agg.Aggregate}(\mathsf{PP}, \mathsf{sk}_{\widetilde{id}}, (\mathsf{ak}, (c_1, c_2, c_3)), (\mathsf{ak}', (c_1', c_2', c_3')))$.

$\quad (\alpha_{\widetilde{id}}, \mathsf{sk}_{\mathsf{IBE}}, \mathsf{sk}_{\mathsf{IBE}}') \leftarrow \mathsf{sk}_{\widetilde{id}}$

$\quad \alpha \leftarrow \mathsf{ak}$

$\quad \alpha' \leftarrow \mathsf{ak}'$

$\quad \kappa \leftarrow \mathcal{E}_{\mathsf{SKE}}.\mathsf{Dec}(\alpha, c_1)$

$\quad \kappa' \leftarrow \mathcal{E}_{\mathsf{SKE}}.\mathsf{Dec}(\alpha', c_1')$

$\quad$ If $\kappa = \bot$ or $\kappa' = \bot$:

$\quad\quad$ Output $\bot$

$\quad (h, v, z) \leftarrow \mathcal{E}_{\mathsf{SKE}}.\mathsf{Dec}(\kappa, c_3)$

$\quad (h', v', z') \leftarrow \mathcal{E}_{\mathsf{SKE}}.\mathsf{Dec}(\kappa', c_3')$

$\quad$ If $h \neq h'$:

$\quad\quad$ Output $\bot$

$\quad s_1, s_2 \leftarrow_\$ \mathbb{Z}_N$

$\quad v'' \leftarrow v * v' * z^{s_1}$

$\quad z'' \leftarrow z^{s_2}$

$\quad \kappa'' \leftarrow \mathcal{E}_{\mathsf{SKE}}.\mathsf{Gen}(1^\lambda)$

$\quad c_1'' \leftarrow \mathcal{E}_{\mathsf{SKE}}.\mathsf{Enc}(\alpha_{\widetilde{id}}, \kappa'')$

$\quad c_2'' \leftarrow \mathcal{E}_k.\mathsf{Enc}(\mathsf{PP}_{\mathsf{IBE}}', h, \kappa'')$

$\quad c_3'' \leftarrow \mathcal{E}_{\mathsf{SKE}}.\mathsf{Enc}(\kappa'', (h'' := h, v'', z''))$

$\quad$ Return $(c_1'', c_2'', c_3'')$

---

### 6.2. F-Type Setting

Now, we turn our attention to the more challenging problem of obtaining aggregation validity together with strong unlinkability in the F-type setting of full anonymity. We observe that we can solve this problem with (identity-based) fully homomorphic encryption (FHE). The idea is to encrypt the hash $h$ with an identity-based FHE scheme to obtain ciphertext $\psi_h$ and place $\psi_h$ in the tuple $(h, v, z)$ instead of $h$. The aggregator can then homomorphically produce encryption of a fresh key under identity $h$ by performing homomorphic evaluation on $\psi_h$. The additional expense of homomorphic evaluation aside, the major prohibitive factor of this approach is the fact that bootstrapping is necessary to achieve unlinkability, and this requires us to make a circular security assumption. Hence we seek to solve the problem in an alternative way, avoiding FHE and bootstrapping.

Instead, we rely on an IND-ID-RCCA secure IBE scheme that is both anonymous and satisfies strong unlinkability with the ability to generate rerandomizable anonymous encryption keys for a particular identity. We make use of our anonymous IBE scheme from the previous section to fulfill our requirements. Recall that this scheme comes with two useful algorithms:

- $\mathsf{GenAnonKey}(\mathsf{PP}, \mathsf{id})$.
- $\mathsf{RerandomizeKey}(\mathsf{PP}, \mathsf{AnK})$.

Given the public parameters and an identity string, Algorithm $\mathsf{GenAnonKey}$ generates an anonymous key $\mathsf{AnK}$, which hides the identity and can be used to encrypt a message for that identity. The second algorithm, $\mathsf{RerandomizeKey}$, given the public parameters and an anonymous key, derives an unrelated anonymous key for the same identity such that no party can link the keys and determine that they are related (i.e., have the same intended recipient). The anonymous key is preprended to every ciphertext generated with it, so, therefore, it is advantageous to rerandomize it, so the ciphertexts are not linked to each other. Algorithm 6 shows how this algorithm is used in our IBAA scheme's aggregation algorithm for the F-type setting. Note that although we do not show it, it is also necessary to slightly modify the encryption and decryption algorithms of our IBAA scheme to accommodate the F-type setting.

---

**Algorithm 6** Our IBAA scheme aggregation algorithm for F-type setting.

---

$\mathsf{Agg.Aggregate}(\mathsf{PP}, \mathsf{sk}_{\widetilde{\mathsf{id}}}, (\mathsf{ak}, \mathsf{ct}), (\mathsf{ak}', \mathsf{ct}')).$

$(\alpha_{\widetilde{\mathsf{id}}}, \mathsf{sk}_{\mathsf{IBE}}, \mathsf{sk}'_{\mathsf{IBE}}) \leftarrow \mathsf{sk}_{\widetilde{\mathsf{id}}}$

$(c_1, c_2 := (\mathsf{AnK}, \psi), c_3) \leftarrow \mathsf{ct}$

$(c'_1, c'_2 := (\mathsf{AnK}', \psi'), c'_3) \leftarrow \mathsf{ct}'$

$\alpha \leftarrow \mathsf{ak}$

$\alpha' \leftarrow \mathsf{ak}'$

$\kappa \leftarrow \mathcal{E}_{\mathsf{SKE}}.\mathsf{Dec}(\alpha, c_1)$

$\kappa' \leftarrow \mathcal{E}_{\mathsf{SKE}}.\mathsf{Dec}(\alpha', c'_1)$

If $\kappa = \bot$ or $\kappa' = \bot$:

    Output $\bot$

$(v, \bar{v}, z) \leftarrow \mathcal{E}_{\mathsf{SKE}}.\mathsf{Dec}(\kappa, c_3)$

$(v', \bar{v}', z') \leftarrow \mathcal{E}_{\mathsf{SKE}}.\mathsf{Dec}(\kappa', c'_3)$

$s_1, s_2, s_3 \leftarrow_{\$} \mathbb{Z}_N$

$v'' \leftarrow v * v' * z^{s_1}$

$\bar{v}'' \leftarrow \bar{v} * \bar{v}' * z^{s_2}$

$z'' \leftarrow z^{s_3}$

$\kappa'' \leftarrow \mathcal{E}_{\mathsf{SKE}}.\mathsf{Gen}(1^\lambda)$

$c''_1 \leftarrow \mathcal{E}_{\mathsf{SKE}}.\mathsf{Enc}(\alpha_{\widetilde{\mathsf{id}}}, \kappa'')$

$\mathsf{AnK}'' \leftarrow \mathsf{RerandomizeKey}(\mathsf{PP}'_{\mathsf{IBE}}, \mathsf{AnK})$

$c''_2 \leftarrow (\mathsf{AnK}'', \mathcal{E}_k.\mathsf{Enc}(\mathsf{PP}'_{\mathsf{IBE}}, \mathsf{AnK}'', \kappa''))$

$c''_3 \leftarrow \mathcal{E}_{\mathsf{SKE}}.\mathsf{Enc}(\kappa'', (v'', \bar{v}'', z''))$

Return $(c''_1, c''_2, c''_3)$

---

**Author Contributions:** Cryptography M.C.; Project supervision H.T. All authors have read and agreed to the published version of the manuscript.

**Funding:** This research recevied funding ADAPT grant number 13/RC/2106_P2 and CONNECT grant number 13/RC/2077_P2.

**Data Availability Statement:** Not applicable.

**Conflicts of Interest:** The authors declare no conflict of interest.

### Abbreviations

The following abbreviations are used in this manuscript:

| | |
|---|---|
| IBE | Identity-Based Encryption |
| TA | Trusted Authority |

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
