# Peer review of "Anonymous Homomorphic IBE with Application to Anonymous Aggregation"

_cryptography, doi:10.3390/cryptography7020022_

Round 1

Reviewer 1 Report

In this paper, the authors presented the construction of an anonymous group-homomorphic IBE scheme that does not sacrifice anonymity to perform homomorphic operations. As the authors signed, all anonymous identity-based encryption schemes that are group homomorphic require knowledge of the identity to compute the homomorphic operation. Thus, new, better methods for performing homomorphic operations are required.

Article sounds very scientific. I suggest it for publication in present form.

Author Response

Please find attached our response

Reviewer 2 Report

I am not an expert in encryption or cybersecurity. But my overall impression after going through the presentation of the paper is that the paper deals with an important problem.

Authors have provided lots of background information to set up the basis of the proposed approach, which might not be easy to read by some readers. Especially, the excessive amount of formulation could distract readers.

Authors have tried to present in detail their proposed approach and subsequent constructions. Yet, some justification should accompany each new proposal to ensure the intuition behind could be clearly understood.

The verification of the proposed methods could be doubtable. I understand many cybersecurity or privacy-related research use theoretical analysis to replace empirical studies. But there is definitely some limitation to the pure theoretic approach. The work would be more convincing if it is tested and verified in a practical scenario.

Author Response

Please find attached our response

Reviewer 3 Report

This paper presents a construction of anonymous homomorphic IBE scheme assuming iO and the hardness of the DBDH problem over rings. And then extends it to build an identity-based anonymous aggregation protocols. The result gives an answer to an open problem. 

My suggestions for the authors are as follows.

1. In page 6, some formulas such as "Pr C_0 (x) = C(x) : C ' iO(C) = 1" should be refined to"Pr [C_(x) = C(x) : C ' ← iO(C) ]= 1" for clarity.

2. In page 7, "is in instance of"should be "is an instance of".

3. In page 10, what K stands for in "MSK := (K, s, MSK_IBE, sk_T)" should be explained.

4. In page 10, Does PKE.Enc(pk_T, b; ρ) mean encrypt ρ using public key  pk_T with randomness b?

5. In page 11, line 400, "Hyrbid 1" should be "Hybrid 1" .

6. In the Proof of Theorem 3, why "Hybrid 2" is missing from the hybrid argument.    

Author Response

Please find attached our response
